# Longitudinal Study on the Effect of Season and Weather on the Behaviour of Domestic Cats (*Felis catus*)

**DOI:** 10.3390/ani15050637

**Published:** 2025-02-22

**Authors:** Michelle Smit, Christopher J. Andrews, Ina Draganova, Rene A. Corner-Thomas, David G. Thomas

**Affiliations:** School of Agriculture and Environment, Massey University, Palmerston North 4410, New Zealand; c.j.andrews@massey.ac.nz (C.J.A.); i.draganova@massey.ac.nz (I.D.); r.corner@massey.ac.nz (R.A.C.-T.)

**Keywords:** feline, domestic cat, research cat, seasonal behaviour, weather

## Abstract

This study explored how seasonal and weather variations influence domestic cat behaviours. Using accelerometer data and a validated machine learning model, eight behaviours—active, eating, grooming, littering, lying, scratching, sitting, and standing—were tracked in seven research cats over 13 weeks throughout the year, alongside concurrent weather data collection. Generalised linear mixed models revealed seasonal differences for eating, grooming, littering, lying, scratching, and sitting but not for active behaviours or standing. A higher temperature humidity wind index and longer daylength increased time spent eating, lying, and standing while reducing time spent active, grooming, littering, and sitting. More rain led to less time grooming and scratching. These findings highlight seasonality in cat behaviours, influenced by weather conditions, and can aid in providing guidance to cat facility managers as to when additional resources could be beneficial, such as brushing cats in times when grooming and scratching are increased.

## 1. Introduction

Any living individual displays at least one behaviour at any given time, which is a response to either internal or external stimuli, or a combination of both [1]. Domestic cats can be exposed to a range of external stimuli that differ based on how they are housed. A recent large-scale study across Europe, North America, Australia and, New Zealand reported that approximately 59% of pet cats had access to the outdoors [2]. In New Zealand, an estimated 83% (~1 million) of pet cats have outdoor access [3], where they are exposed to seasonal and meteorological changes (i.e., weather patterns). To date, little is known about how these changes might affect the behaviour of domestic cats.

Seasons are characterised by differences in daylength and weather. The natural light–dark cycle plays a key role in the timing of peak activity of organisms [4,5], including domestic cats [6,7,8]. Daylength also plays a role in seasonal activity, with [9] reporting that indoor research cats, housed under natural light conditions and constant temperature and relative humidity, travelled greater distances in both spring and autumn than in winter. Other studies also reported differences in activity between seasons, although this differed between housing conditions. Free-roaming cats have been shown to be more active in spring and summer than in autumn and winter [10,11,12]. These studies, however, did not distinguish between pet, stray, or feral cats, and did not attempt to identify specific behaviours. Horn et al. [13] compared the activity of unowned free-roaming cats to that of free-roaming pet cats and found contrasting results. Unowned cats were most active in autumn and winter, while pet cats were more active in spring and autumn. They attributed these differences to higher energy demands during the colder months in the unowned cats, while pet cats appeared to prioritise comfort, avoiding extreme hot and cold conditions. While seasonal data can provide valuable insights, utilised alone it does not specifically indicate which weather variable might play a role in behavioural changes.

In free-roaming cats, activity has been reported to be positively and negatively correlated with temperature and rain, respectively [10,11,12]. Konecny et al. [14] reported the lowest activity in feral cats was around midday when the ambient temperature was at its highest. While Izawa [15] also reported a negative correlation between temperature and activity during the day, this correlation was positive during the night. A questionnaire-based study, including both indoor-only and free-roaming pet cats, found that extreme weather events affected specific behaviours of pet cats [16]. Sudden decreases in temperature were associated with an increase in eating behaviour, whereas intake decreased in hot weather (temperature not specified). There was, however, a lack of detail on what constituted excessively hot temperatures, and owner-reported data are subjective [16].

Current knowledge of the effects of season and weather changes on cat behaviour have primarily come from activity studies using visual observation, radio tracking, Global Positioning Systems (GPS), or accelerometers. While these studies can provide valuable insights into activity patterns, they do not capture the full range of behaviours that cats display. Traditionally, for behavioural studies, researchers have relied on direct observation of the animal(s) and manual recording of behaviours using an ethogram [17,18]. Though effective, direct observation is labour intensive, limited by the ability of the observer to continuously monitor animals for long periods or in challenging environments, and is subjective [17,18]. The emergence of video recording devices has allowed more detailed behaviour capture and analysis to occur, with the ability to pause and review footage multiple times, thereby limiting the effect of observer fatigue [18]. In addition, video recording, when operated from a distance or remotely, can limit the potential effect of the presence of an observer on animal behaviour [18]. Scoring behaviour from video recordings, however, remains labour intensive. In addition, several studies have reported high interindividual variation in activity among domestic cats, despite being housed under the same conditions [19,20,21]. The use of high-frequency accelerometer data in combination with machine learning (ML) has emerged as an alternative, less labour-intensive tool to classify animal behaviours.

Several studies have now validated ML models to continuously and objectively classify domestic cat behaviour from acceleration data, providing more comprehensive data than traditional observation methods [22,23,24,25]. Accelerometers and ML offer great potential in studying behavioural responses to external stimuli, such as seasonal and weather-related changes. A previous study examined the effects of season and housing conditions on eight behaviours of pet cats, with behaviour being classified using accelerometers and machine learning (ML) [26]. In this study, pet cats with outdoor access showed lower levels of active behaviours (walking and trotting) in winter than summer, whereas no seasonal differences were observed among indoor-only pet cats. The remaining seven behaviours (eating, grooming, littering, lying, scratching, sitting, and standing) were not affected by season or housing but were influenced by other environmental factors, including the presence of children and other cats. Previously validated ML models have been used successfully in other animal species for behavioural studies, for example, chipmunks (*Tamias* spp.) [27], oystercatchers (*Haematopus ostralegus*) [28], and pumas (*Puma concolor*) [29]. While these studies did not specifically examine the effects of environmental factors such as temperature, humidity, and rainfall on animal behaviour, they illustrated the power of accelerometery and machine learning for studying animal behaviour. The current study aimed to investigate seasonal domestic cat behavioural patterns, quantified using accelerometer data and a model previously validated by the same authors [24], and the influence of weather patterns on them, in a controlled environment.

## 2. Materials and Methods

This study was conducted at the Massey University Centre for Feline Nutrition, Palmerston North, New Zealand (latitude 40°23′ S, longitude 175°36′ E). The study was approved by the Massey University Animal Ethics Committee (MUAEC 22/23).

### 2.1. Animals and Housing

Eight healthy, desexed male (n = 4) and desexed female (n = 4) domestic shorthair cats were group housed in a single semi-outdoor colony cage for a year (Figure 1). At the start of the trial, the average ± SD age of the cats was 4.2 ± 0.8 years, and bodyweight was 4.2 ± 0.9 kg. The bodyweight of every cat was measured on a weekly basis throughout the trial. Cats were fed a complete and balanced [30] commercial canned diet (Heinz Wattie’s Ltd., Hastings, New Zealand), and had ad libitum access to water for the duration of the study.

### 2.2. Accelerometer Data Collection

ActiGraph™ wGTX-BT accelerometers (weighing 19 g and measuring 33 mm × 46 mm × 15; dynamic range ± 8 g; ActiGraph^®^ Pensacola, FL, USA) were positioned ventrally on a collar (Figure 2a), with the orientation of the x-, y-, and z-axes laterally, dorso-ventrally, and cranio-caudally, respectively (Figure 2b). Acceleration data were sampled at a frequency of 30 Hz (defined as the raw acceleration data) and then downloaded and exported into csv files using ActiLife software (version 6.13.4; ActiGraph™, Pensacola, FL, USA). Raw acceleration data were feature engineered to and summarised into 1 s epochs to obtain the 32 variables required by the behaviour classification model as previously described in [24]. Using the feature-engineered data, the behaviour of the cats was classified using a collar-based model previously validated by [24]. The model classifies behaviour for every 1 s datapoint and had an overall accuracy of ~73% and F1-score of ~68% and classified eight different behaviours: active (walking and trotting), eating, grooming, littering, lying, scratching, sitting, and standing (see Table 1 for behaviour descriptions). This combination of behaviours was selected due to their performance in the ML model and potential medical relevance (e.g., scratching for dermatological problems) as described in [24]. Accelerometer data were collected for seven consecutive days, every four weeks for a year, starting in March 2023, and ending in February 2024 (Figure 3), capturing a total of 91 days of accelerometer data per cat. Events that deviated from normal were noted, such as a social disturbance resulting from a new individual being introduced to a pen, or groups of students visiting for practical classes.

### 2.3. Weather Data Collection

A Vantage Pro2™ ISS weather station (Davis Instruments, Hayward, CA, USA) located at the Centre for Feline Nutrition (−40.390344, 175.615829) collected hourly weather data continuously from 1 March 2023 till 29 February 2024. The weather station was connected to a solar panelled gateway EnviroMonitor System (Davis Instruments, Hayward CA, USA), which sent hourly weather data via cellular connection to https://www.weatherlink.com/, from which the hourly weather data were downloaded. Seven weather variables were selected: average temperature (°C), minimum and maximum temperature (°C), average relative humidity (%), average temperature, humidity and windchill index (THW, °C), average wind speed (m/s), and total rainfall (mm; Table 2). Four seasons were recognised: autumn (March–May), winter (June–August), spring (September–November), and summer (December–February).

The R package ‘suncalc’ [38] was used to obtain daily data on the exact times of light phases based on the longitude and latitude coordinates of the weather station (−40.390344, 175.615829). With these data, daylength was determined.

### 2.4. Statistical Analyses

All data computation and statistical analyses were carried out using RStudio version 4.1.1 [39]. Statistical significance was defined as *p* ≤ 0.05, with a trend being 0.05 > *p* < 0.10. All results are presented as the mean ± standard error of the mean (SEM).

#### 2.4.1. Seasonal Differences

Following behaviour classification, the data were cleaned by removing acceleration data from the times each cat was not wearing the collar. Seasons were assigned to the individual datapoints in the dataset based on the calendar date. For each season, the proportion of time each cat was classified as exhibiting each behaviour was determined. Outliers were identified using boxplots and were removed if they occurred during a noted event or disturbance. Generalised linear mixed models (GLMMs), using the R package ‘glmmTMB’ [40], with a beta distribution and logit link to account for the proportional nature of the data, were used to statistically test the proportional behavioural data. Three GLMMs were created for each behaviour: (1) a simple model with no predictors, (2) an intermediate model with only cat as a random effect, and (3) a full model that included cat as a random effect and the season as a fixed effect as in Equation (1):(1)$$logit\left(\mu_{ij}\right)=\beta_{0}+\beta_{1}X_{ij}+u_{j}$$
where $\mu_{ij}$ is the expected proportion of time cat *j* spends on the behaviour in observation *i*, $\beta_{0}$ is the intercept, $\beta_{1}$ is the fixed effect of season, $X_{ij}$ is the categorical variable indication season for observation *i* for cat *j*, and $u_{j}\sim N(0,\;\sigma^{2})$ is the random intercept for cat *j* to account for individual differences.

The three models were compared with an ANOVA to determine which factors improved the model, and the marginal and conditional R^2^ of the full models were determined. The marginal R^2^ value is the variance explained by only the fixed effect, whereas the conditional R^2^ value is the variance explained by both the random and fixed effects. Bodyweights were averaged over the seasons for each cat and analysed using a linear mixed-effects model to determine seasonal differences. To determine differences between the seasons, pairwise comparisons of estimated marginal means were conducted for each behaviour and cat bodyweight using the R package ‘emmeans’ [41]. A Tukey adjustment was used to correct for multiple pairwise comparisons.

#### 2.4.2. Effect of Weather Variables on Behaviour

Daily proportions for each behaviour were calculated using the cleaned dataset, and daily averages for temperature (including minimum and maximum), relative humidity, THW index, and wind speed, and the sum of rainfall were calculated. These daily averages were merged with daily behavioural proportions based on date, and the effect of weather conditions were statistically tested using a GLMM with a beta distribution and logit link. Similar to the GLMM testing above, three GLMMs were performed, with the same simple and middle models. The full model included cat as a random effect and weather conditions as fixed effects. The full model followed the same mathematical formulation and assumptions as in Equation (1) but with the fixed effects being weather variables.

The marginal and conditional R^2^ values for each model were determined. The inclusion of weather conditions as fixed effects was determined based on their correlation coefficients, which were interpreted as defined by [42]. Weather data were scaled using the scale function in R prior to statistical modelling.

## 3. Results

Data collection originally scheduled for calendar week 43 (trial week 9) was postponed by a week until week 44 due to illness of the researcher. One of the cats was humanely euthanised due to an illness unrelated to the study; therefore, only data from seven cats were included in the data analyses. One week of data was not collected for two cats, one in trial week 9 and one in trial week 12, due to animal illness.

A small number of outliers were identified in the dataset (Figure 4). Most identified outliers for grooming and scratching were found to belong to one cat: Cho. She was diagnosed with seasonal allergies, and therefore, her grooming and scratching data were excluded from the data analyses. Large variations in many behaviours were observed among all cats in trial week 5 when compared to other trial weeks (Appendix A). In that week, one cat was placed back into an incorrect pen, which appeared to impact both its behaviour and that of the other cats in the study. For this reason, the behavioural data of trial week 5 were removed from the dataset for all cats and excluded from data analyses. The previously mentioned outliers were removed as they could have introduced confounding effects. Large variations in behaviour were observed for Nimbus in trial week 10, but no events or disturbances were reported for that week, and, therefore, these data were not removed (Appendix A). Table 3 shows the seasonal and total amount of data included per cat in the final data analyses, following the removal of when cats were not wearing the collar and outliers as defined previously.

### 3.1. Seasonal Differences

#### 3.1.1. Bodyweight

Cats were weighed weekly (Figure 5), and their weights averaged per season of the year. Six cats were included in the analysis due to weight loss in one cat (Mrs. Norris). No significant differences in average bodyweight were identified among seasons (4.2 ± 0.4 kg; *p* > 0.05).

#### 3.1.2. Behaviours

Inclusion of cat as a random effect improved the active behaviours GLMM (*p* = 0.003) but not the GLMM of eating, grooming, littering, lying, scratching, sitting, and standing (*p* > 0.05; Table 4). Inclusion of season as a fixed effect improved the GLMM model for eating (*p* < 0.001), grooming (*p* < 0.001), littering (*p* = 0.037), lying (*p* < 0.001), scratching (*p* < 0.001), and sitting (*p* < 0.001) but not for active behaviours and standing (*p* > 0.05; Table 4).

Cats were observed to spend the most time eating in spring (11.34 ± 0.48%), followed by summer (9.52 ± 0.48%) and winter (7.73 ± 0.83%), and the least in autumn (1.73 ± 0.42%; Figure 6a). Time spent eating was lower in autumn compared to the three other seasons (*p* < 0.001) and was lower in winter than in both spring (*p* < 0.001) and summer (*p* = 0.035). A trend was found for the difference between spring and summer (*p* = 0.057).

Cats groomed most in autumn (6.83 ± 0.47%), followed by winter (4.57 ± 0.70%), spring (4.46 ± 0.56%), and summer (4.04 ± 0.47%; Figure 6b). Time spent on grooming was higher in autumn compared to all the other three seasons (*p* < 0.001). A trend was found for higher levels in winter compared to summer (*p* = 0.083) and no differences were found between spring and either winter or summer (*p* > 0.05).

Cats littered most in autumn (0.046 ± 0.012%), followed by summer (0.022 ± 0.004%), winter (0.017 ± 0.004%), and spring (0.013 ± 0.009%; Figure 6c). Time spent littering was higher in autumn compared to winter (*p* = 0.022), spring (*p* = 0.002), and summer (*p* = 0.021). No differences were found between winter, spring, and summer (*p* > 0.05).

Cats spent most time lying in summer (60.33 ± 3.92%), followed by spring (58.29 ± 3.37%), winter (52.88 ± 3.52%), and autumn (28.68 ± 3.93%; Figure 6d). Time spent on lying was lowest in autumn compared to all other seasons (*p* < 0.001). A trend was found for the difference between winter and summer (*p* = 0.076), and no difference was found between spring and summer (*p* > 0.05).

Cats scratched most in autumn (0.16 ± 0.03%), followed by winter (0.05 ± 0.00%), summer (0.04 ± 0.02%), and spring (0.03 ± 0.01%; Figure 6e). Cats spent the most time scratching in autumn compared to the other three seasons (*p* < 0.001). Scratching was higher in winter compared to both spring (*p* = 0.008) and summer (*p* = 0.014). No difference was found between spring and summer (*p* > 0.05).

Cats sat most in autumn (50.77 ± 3.73%), followed by winter (20.10 ± 1.14%), summer (14.35 ± 0.97%), and spring (11.98 ± 1.61%; Figure 6f). Cats spent more time sitting in autumn compared to the other three seasons (*p* < 0.001). Cats spent more time sitting in winter compared to both spring (*p* < 0.001) and summer (*p* = 0.013) and no difference was found between spring and summer (*p* > 0.05).

Despite not being significantly different, cats were observed to display active behaviours (walking and trotting) between 1.56 ± 0.54% (in summer) and 2.47 ± 0.27% (in spring; Figure 6g) of their time, and cats stood between 9.15 ± 0.53% (in autumn) and 12.20 ± 1.38% (in winter) of their time (Figure 6h).

### 3.2. Effect of Weather Variables on Behaviours

Data on weather variations can be found in Appendix A. Correlation coefficients between weather conditions and daylength are presented in Table 5. A very high correlation was found between the THW index and temperature (0.99), and a high correlation was found between temperature and relative humidity (−0.71). Moderate correlations were found between the THW index and relative humidity (−0.64), daylength and temperature (0.51), and daylength and the THW index (0.50). A low correlation was found between wind speed and relative humidity (−0.46). All other correlations were negligible. Due to the moderate correlation between daylength and the THW index, these were included in the model as an interaction. Thus, ‘THW × daylength’ and rainfall were included in the GLMM.

For all behaviours, the inclusion of ‘THW index × daylength’ and rainfall as fixed factors significantly improved the model (*p* < 0.001; Table 6). Inclusion of cat as a random effect improved the model for all behaviours (*p* < 0.001), except for sitting (*p* > 0.05; Table 6). The variance explained by the model differed between behaviours, with the most variance explained for lying (R^2^_conditional_ = 0.68), sitting (R^2^_conditional_ = 0.61), and eating (R^2^_conditional_ = 0.57; Table 6). The model explained < 0.50 of the variance for each of the other behaviours. The ‘THW index × daylength’ interaction had a significant effect on all behaviours (*p* < 0.05; Table 6). Both THW index and daylength were negatively correlated with the time spent in active behaviours, grooming, littering, scratching, and sitting but positively correlated with eating, lying, and standing. Rainfall negatively affected time spent grooming (*p* = 0.023) and scratching (*p* = 0.037), and a negative trend was found for littering (*p* = 0.053; Table 6).

## 4. Discussion

Longitudinal studies of animal behaviour have traditionally been challenging due to their labour-intensive observational methods. Developing ML models to classify cat behaviours from acceleration data has significantly reduced this and allowed for detailed longitudinal behavioural studies. The semi-outdoor living conditions of the cats at the Massey University Centre for Feline Nutrition, which exposes them to natural light and weather conditions, provided an opportunity for a longitudinal study into the effects of season and weather on cat behaviour.

In healthy individuals, changes in bodyweight are the result of an imbalance in energy requirements and energy intake [44]. In this study, no seasonal differences in bodyweight were observed, suggesting there was a balance between energy intake and requirements throughout the year. By contrast, Bermingham et al. [45] observed a seasonal change in bodyweight among cats housed in the same research facility, with bodyweight increasing in the months leading up to winter and decreasing in spring.

Data on seasonal changes in the energy requirements of domestic cats are scarce and contradictory. Kappen et al. [46] reported that indoor-housed research cats required a lower energy intake during short-day (winter) compared to long-day conditions (summer) to maintain their bodyweight. Similarly, Bermingham et al. [47] reported lower energy requirements per kg of bodyweight in winter than in summer for older semi-outdoor-housed research cats (~10 years) but not in younger cats (~3 years) housed either indoors or semi-outdoors. When energy requirements were expressed per kilogram of lean body mass, which closely reflects the amount of metabolically active tissue [44], seasonal differences in the energy requirements were no longer observed in the older cats [47]. Among cats fed ad libitum, bodyweight did not differ between winter and summer, regardless of age and housing condition [47], suggesting energy requirements did not differ seasonally. Conversely, Serisier et al. [48] reported that energy intake in research cats, housed both indoors only and indoors/outdoors and fed ad libitum, was highest in in late autumn and winter. Bodyweight, however, did not change significantly throughout the seasons, suggesting a change in energy requirement with the seasons was driven by maintaining body temperature or changes in physical activity (PA).

Kappen et al. [46] observed a lower PA (measured with an accelerometer) during short-day conditions than long-day conditions, which could have contributed to the lower energy requirements observed during the short-day conditions. Similarly, in the current study, no differences in the amount of time spent on active behaviours were observed between the seasons for cats fed ad libitum, which could explain their stable bodyweights throughout the year. Neither [47] nor [48] measured PA, although [48] argued that the PA of cats in their study remained constant throughout the study, as the housing conditions and periods for free and play activity remained constant. Differences in housing that could have resulted in different energy requirements could be an explanation for the differences in seasonal bodyweight change between the current study and those observed by [45], but this is unlikely as they were housed similarly.

Another factor that can contribute to changes in energy requirements is thermoregulation. Energy requirements increase with decreasing ambient temperatures in order for animals to maintain their body temperature [44]. In evolutionary terms, increasing food intake in anticipation of colder weather would be advantageous for survival, helping to build energy reserves for times when food is scarce. Serisier et al. [48] found that ambient temperature and daylength impacted food intake. In autumn, cats also grow a denser fur coat in anticipation of colder weather, driven by a decrease in daylength and ambient temperatures [49,50,51]; this requires additional energy expenditure. The discrepancies reported for seasonal bodyweight changes in domestic cats could be due to different climatic conditions across studies, or individual cat variability. When averaged, no seasonal trends in bodyweight were apparent, but some individuals followed the pattern observed by [45], whereas the opposite was seen for others. Reasons for the differences between individuals in the present study are unknown but could be due to differences in individual activity levels or their position in the hierarchy (in a colony setting) and therefore access to food. Hierarchy was not determined in this study, and given the contradictory results and limited data, more research into seasonal fluctuations in energy requirements and bodyweight is warranted.

The weather conditions in the current study, particularly the interaction of the THW index and daylength, affected cat behaviour. Similar correlations between temperature, rainfall, and daylength and domestic cat activity have been reported, with rainfall shown to reduce cat activity [10,11,15,52]; however, rainfall had no effect on the time cats spent on active behaviours in the current study. This difference in results may be due to the cats’ living conditions, given that this study utilised research cats that had limited space to roam, whereas other published studies focused on free-roaming domestic cats. A previous study in the same colony as the current study also found a negative relationship between rain and PA measured with an accelerometer and expressed as counts [53]. It is likely this negative relationship was the result of a decrease in grooming and scratching behaviour, both of which were negatively affected by rainfall in the current study, as those have been reported to lead to high activity counts by triggering the accelerometer [19].

While the present study found a negative relationship between the interaction of the THW index and daylength on time spent on active behaviours, others have reported a positive relationship [10,11]. Several factors could explain this difference. Izawa [15], for example, reported that the relationship between cat activity and temperature was positive at night but negative during the day. Environmental factors (time of day, temperature, relative humidity, and rain) have been reported to account for more variance in overall activity in a population with a larger proportion of feral cats (32.6%) compared to free-roaming pet cats (14.9%), suggesting that feral cats are more sensitive to environmental conditions [11]. In the current study, the fixed effects of THW index × daylength and rainfall explained only ~3% of the variation in time spent showing active behaviours, but this increased to ~46% when individual differences between cats were accounted for. These results suggest there is a complex relationship between activity, the weather, and living conditions, which may vary between individual cats. Additionally, weather conditions can change rapidly, even within an hour. These short-term fluctuations, however, were not captured in the current study, as data were aggregated daily rather than hourly. Future research could explore the relationship between domestic cat behaviour and weather conditions at an hourly resolution to uncover the fine-scale dynamics.

Time spent grooming, scratching, and sitting peaked in autumn, which was the wettest season recorded during the study. Interestingly, despite this, a negative relationship was observed between rainfall and time spent on these behaviours. Although this relationship was significant, the fixed weather effects accounted for only ~16% and ~21% of the variation in the time spent on grooming and scratching, respectively. This suggests that other factors, such as the seasonal rhythm of hair growth and shedding, may also have played a role. It might be hypothesised that grooming and scratching would increase as cats shed their winter coats in preparation for warmer weather, which typically occurs in spring and summer [51]. The peak in time spent grooming and scratching aligns with a previously reported peak in daily hair growth [50,51] and onset of hair follicle inactivity, which reduces hair loss [49,54]. These results suggest that an increase in coat density during autumn increases grooming and scratching behaviours. Grooming and scratching are often performed in a seated position (personal observation), so the peak in sitting behaviour which occurred in autumn was therefore likely linked to this peak in grooming and scratching in this season.

The R^2^ values in the current study indicated that there were more variables influencing cat behaviour than just the weather. This is perhaps not surprising given that the definition of behaviour, is “the internally coordinated response (actions or inactions) of whole organisms (individuals or groups) to internal and/or external stimuli” [1]. Another study in pet cats by the same authors found that behaviours were significantly affected by housing (indoor vs. outdoor), number of cats in the household, and the presence of children (<18 years) [26]. All cats in the current study were desexed; therefore, seasonal hormones relating to reproduction did not play a role in behaviour changes. It is likely that entire cats would show seasonal differences as Izawa [15] reported that males active in courtship and mating spent more time displaying active behaviours, and less on feeding and resting, than those not showing these behaviours. The reproductive cycle of queens is affected by daylength, with queens generally cycling during spring and summer when days are longer [55,56]. Breeding, however, can occur year round depending on the animal’s living conditions and latitude [55,56]. Sexual status and cyclicity is thus an important factor to consider in behavioural studies and care should be taken when extrapolating the results from the current study to other individuals that might live under different circumstances.

A challenge in the current study was determining which data points were true outliers and whether to remove them or not. Some outliers were easily identified; for example, a cat was excluded as its grooming and scratching behaviour consistently appeared as outliers and was the result of a seasonal allergy. Additionally, individual boxplots showed large variances in behaviours for all cats during one trial week. This week coincided with the temporary absence of one cat for nearly a full day and its subsequent reintroduction into the colony cage. The increased variance in time spent on behaviours suggests that this event disrupted the group dynamics and affected the behaviour of all cats. Changes in routine, such as relocation or the reintroduction of an absent individual, have been identified as stressors in domestic cats [57,58]. These outliers, which had clear explanations, were excluded from the analyses as they could have introduced confounding effects. Other outliers, however, lacked such explanations. For example, one cat showed a large variation in several behaviours in one trial week (week 5), with no similar patterns observed in the other cats. As no events were recorded that could explain the large variation, the data were retained. Future behavioural studies should clearly define criteria for identifying and handling data to ensure consistency in data analyses.

## 5. Conclusions

This study showed that ML models paired with accelerometer data are powerful tools for conducting detailed, longitudinal studies on cat behaviour. Weather conditions, particularly the interaction between the THW index and daylength, significantly affected cat behaviour, although individual differences between cats also accounted for a large proportion of the variance. Seasonal shifts in grooming and scratching appeared to be linked to seasonal hair growth cycles rather than environmental factors such as rainfall. In addition, this study provided valuable insights into how disruptions, such as the reintroduction of an absent cat, can disrupt behaviour patterns. Further research is needed into the relationship between energy intake, weather conditions, and activity in domestic cats. This study also showed the importance of refining the criteria for identifying outliers and understanding their causes, which will increase the reliability of behavioural studies.

These findings underline the importance of considering both environmental and individual factors when studying domestic cat behaviour, and although this study looked at the effect of weather variables, other environmental factors should be considered, such as different housing and social conditions. Future studies can apply the method to classify domestic cat behaviour in other settings.

## Figures and Tables

**Figure 1 animals-15-00637-f001:**
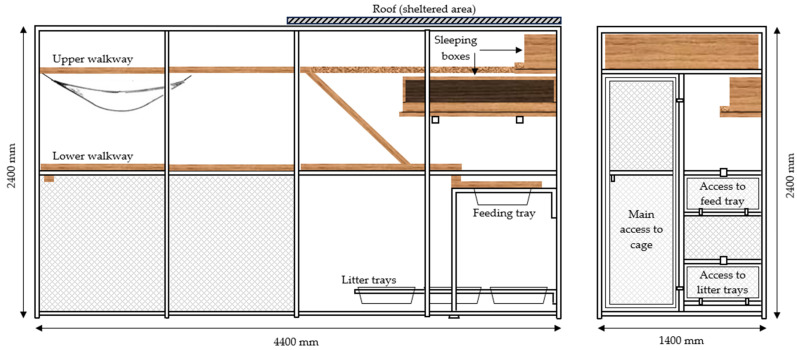
Colony cages as seen from the side (**left**) and front (**right**), measuring 1400 × 2400 × 4400 mm. (Figure first published in [24]).

**Figure 2 animals-15-00637-f002:**
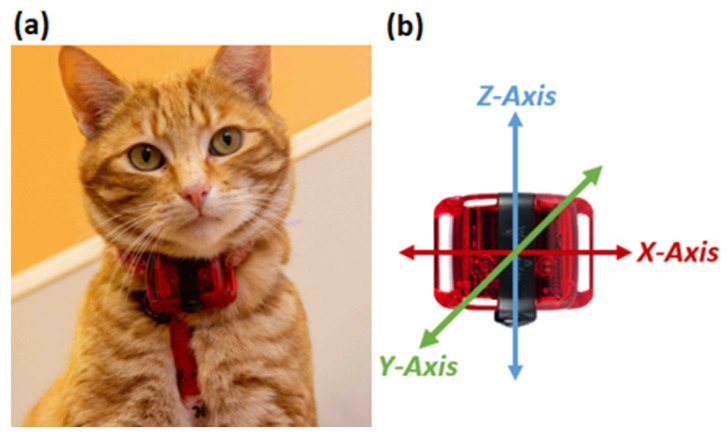
(**a**) Placement and (**b**) orientation of the ActiGraph™ wGT3X-BT accelerometer on a collar. (Figure first published in [24]).

**Figure 3 animals-15-00637-f003:**
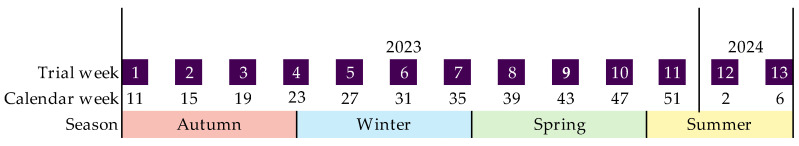
Timeline of data collection, including calendar year and week, trial week, and the season (red = autumn, blue = winter, green = spring, yellow = summer).

**Figure 4 animals-15-00637-f004:**
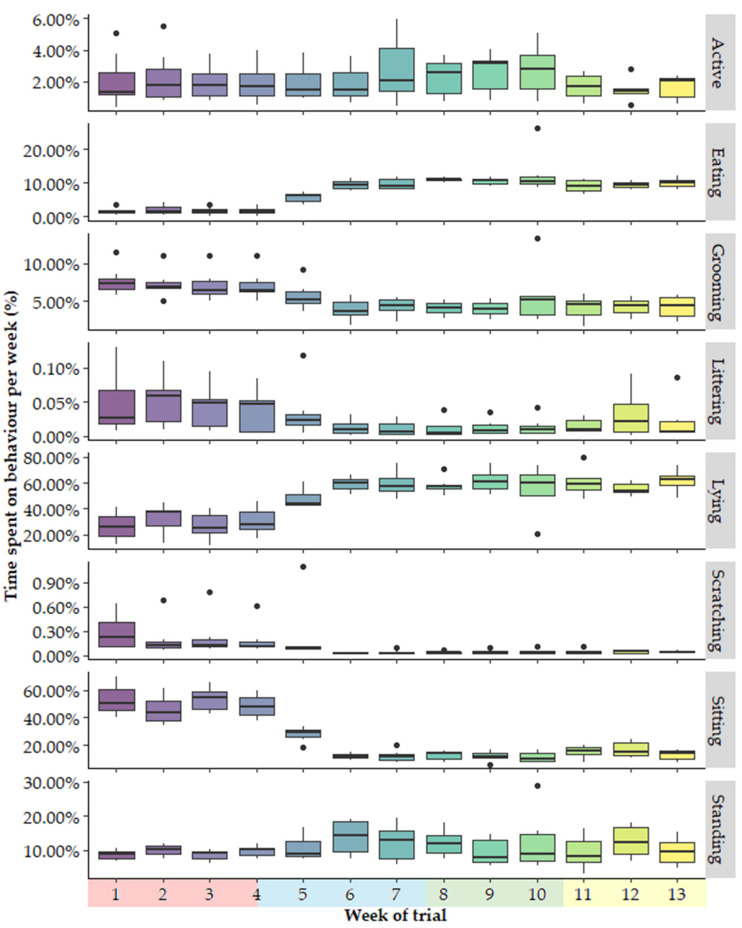
Boxplots of weekly proportional behaviour data. ● indicate outliers and seasons are indicated along the x-axis by colours: red = autumn, blue = winter, green = spring, yellow = summer.

**Figure 5 animals-15-00637-f005:**
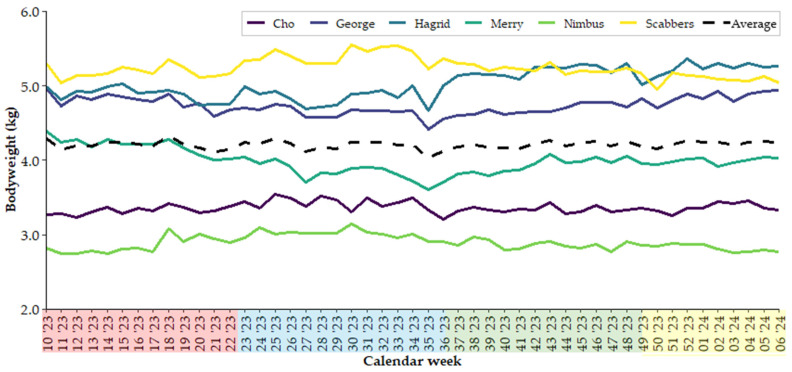
Weekly bodyweight (kg), expressed per calendar week, of all individual cats included in the study. Seasons are indicated along the x-axis by colours: red = autumn, blue = winter, green = spring, yellow = summer.

**Figure 6 animals-15-00637-f006:**
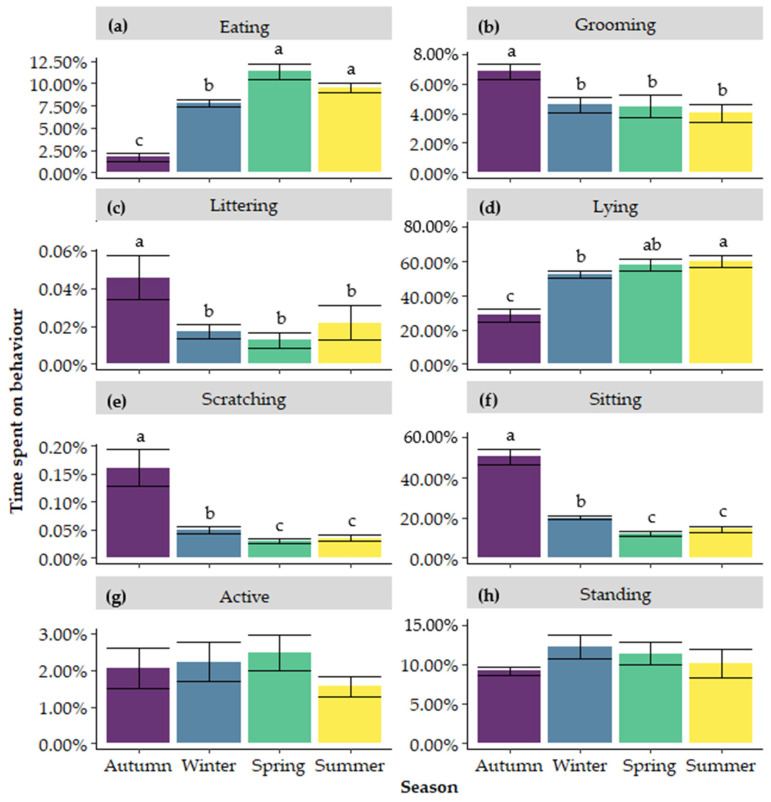
Average time spent (**a**) eating, (**b**) grooming, (**c**) littering, (**d**) lying, (**e**) scratching, (**f**) sitting, (**g**) active, and (**h**) standing in autumn, winter, spring, and summer. ^a–c^ Different superscripts indicate significant differences (*p* ≤ 0.05).

**Table 1 animals-15-00637-t001:** Description of behaviours as identified by the machine learning model. (Definitions first described in [24] and adapted from [31], unless otherwise stated).

Behaviour	Description
Active (walking and trotting)	Walking: forward locomotion at a slow, four-beat and symmetric gait with limbs moving sequentially. Includes slow walk (three or four feet in contact with the ground at any time) and fast walk (two or three feet in contact with the ground at any time). Slowest gait. Trotting: Forward locomotion with a swift, two-beat and symmetric gait. Body is supported by two diagonal legs during contact with ground. Intermediate gait [32,33].
Eating	Cat ingests food (or other edible substances) by means of chewing with the teeth and swallowing.
Grooming	Cat cleans itself by licking, scratching, biting, or chewing the fur on its body. May also include licking of a front paw and wiping it over the head.
Littering	Cat urinates or defecates.
Lying	Cat’s body is in contact with the ground in a horizontal position, including on its side, back, belly, or curled in a circular formation.
Scratching	Cat scratches its body using the claws of its hind feet.
Sitting	Cat is in an upright position, with the hind legs flexed and resting on the ground, while front legs are extended and straight.
Standing	Cat is in an upright position and immobile, with all four paws on the ground and legs extended, supporting the body.

**Table 2 animals-15-00637-t002:** Units and definitions of selected weather variables.

Weather Variable	Unit	Definition [34]
Temperature	°C	Average temperature over the 60 min period.
Minimum temperature	°C	Minimum temperature recorded over the 60 min period.
Maximum temperature	°C	Maximum temperature recorded over the 60 min period.
Relative humidity	%	Average saturation of the air with water at its current temperature over the 60 min period.
THW index	°C	Average calculated temperature per 60 min period that takes temperature, humidity, the heating effects of sunshine, and cooling effects of the wind (wind chill) into account to determine what it feels like in the shade. ^1^
Rainfall	mm	The total rainfall recorded during the 60 min period
Wind speed	m/s	The average wind speed recorded during the 60 min period.

^1^ THW index = heat index − (1.072 × wind speed) [35], with heat index determined based on [36,37].

**Table 3 animals-15-00637-t003:** Seasonal and total amount of days, hours, minutes, and seconds included per cat in the final data analyses.

	Autumn	Winter	Spring	Summer	Total
Cho ^1^	20 days 00:41:13	19 days 00:24:24	20 days 00:07:44	17 days 00:01:42	77 days 00:15:03
George	23 days 00:00:00	20 days 00:43:59	21 days 00:00:00	18 days 00:14:59	82 days 00:58:58
Hagrid	23 days 00:00:00	20 days 00:00:00	21 days 00:00:00	17 days 00:17:59	83 days 00:17:59
Merry	23 days 00:00:00	20 days 00:00:00	21 days 00:00:00	17 days 00:44:59	82 days 00:44:59
Mrs. Norris	23 days 00:00:00	20 days 00:00:00	14 days 00:00:00	18 days 00:00:00	76 days 00:00:00
Nimbus	23 days 00:00:00	20 days 00:00:00	20 days 00:34:59	18 days 00:00:00	83 days 00:34:59
Scabbers	23 days 00:00:00	13 days 00:00:00	21 days 00:00:00	18 days 00:00:00	76 days 00:00:00

^1^ Seconds identified as grooming or scratching behaviour were removed before calculating total days, hours, minutes, and seconds of data included in data analyses.

**Table 4 animals-15-00637-t004:** Marginal and conditional R-squared (R^2^) values for each behaviour.

Behaviour	R^2^	
	Marginal ^1^	Conditional ^1^
Active	0.08	0.66 *
Eating	0.95 *	0.95
Grooming	0.41 *	0.81
Littering	0.25 *	0.38
Lying	0.88 *	0.93
Scratching	0.60 *	0.93
Sitting	0.94 *	0.94
Standing	0.13	0.50

^1^ Marginal R^2^ = variance explained by the fixed effects; conditional variance = variance explained by both the fixed and random effects [43]. * Indicates significant improvement of the model (*p* < 0.001).

**Table 5 animals-15-00637-t005:** Correlation matrix with correlation coefficients of weather conditions and daylength.

	Temperature	Relative Humidity	Wind Speed	THW Index	Rainfall	Daylength
Temperature	1.00					
Relative humidity	−0.71	1.00				
Wind speed	0.27	−0.46	1.00			
THW index	0.99	−0.64	0.15	1.00		
Rainfall	−0.04	0.16	0.08	−0.05	1.00	
Daylength	0.51	−0.36	0.13	0.50	−0.04	1.00

**Table 6 animals-15-00637-t006:** R-squared (R^2^), estimates, and *p*-values for the effect of the THW index, daylength, and rainfall on cat behaviour.

Behaviour	R-Squared		THW Index × Daylength		Rainfall	
	Marginal ^1^	Conditional ^1^	Estimate ^2^	*p*-Value	Estimate ^2^	*p*-Value
Active	0.03 *	0.46 ^†^	−0.079	<0.001	−0.018	0.418
Eating	0.52 *	0.57 ^†^	0.264	<0.001	0.001	0.959
Grooming	0.16 *	0.44 ^†^	−0.087	<0.001	−0.043	0.023
Littering	0.05 *	0.16 ^†^	−0.110	0.004	−0.079	0.053
Lying	0.58 *	0.68 ^†^	0.227	<0.001	−0.034	0.225
Scratching	0.21 *	0.25 ^†^	−0.173	<0.001	−0.082	0.037
Sitting	0.60 *	0.61	−0.290	<0.001	−0.032	0.345
Standing	0.06 *	0.30 ^†^	0.097	<0.001	0.010	0.598

^1^ Marginal R^2^ = variance explained by the fixed effects; conditional = variance explained by both the fixed and random effects [43]. ^2^ Estimate based on the log odds ratio scale. * Indicates inclusion of fixed effects significantly improved the model (*p* < 0.001). ^†^ Indicates inclusion of the random cat effect significantly improved the model (*p* < 0.001).

## Data Availability

The datasets presented in this article are not readily available because the dataset is too large. Requests to access the datasets should be directed to m.smit@massey.ac.nz.

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
