# Peer review of "Longitudinal Study on the Effect of Season and Weather on the Behaviour of Domestic Cats (Felis catus)"

_animals, 2025, doi:10.3390/ani15050637_

Round 1

Reviewer 1 Report

Comments and Suggestions for Authors

I was happy to have the opportunity to review your work. I found it very interesting. I did my best to foster an improvement of the quality. This study reports the effects of seasonal and weather changes on cat behavior using accelerometer data and a machine learning model. The introduction and methodology are appropriately described and presented in a very detailed manner. The paper provides very interesting data, but it still needs a considerable revision to be acceptable for Animals. In particular, a more logical explanation of the effects of both season and weather, as well as a clearer presentation of the results in line with the main topic, is necessary. Please refer to the attached PDF for specific comments that need to be addressed.

Author Response

I was happy to have the opportunity to review your work. I found it very interesting. I did my best to foster an improvement of the quality. This study reports the effects of seasonal and weather changes on cat behavior using accelerometer data and a machine learning model. The introduction and methodology are appropriately described and presented in a very detailed manner. The paper provides very interesting data, but it still needs a considerable revision to be acceptable for Animals. In particular, a more logical explanation of the effects of both season and weather, as well as a clearer presentation of the results in line with the main topic, is necessary. Please refer to the attached PDF for specific comments that need to be addressed.

Comment 1: Including statistical values in the Simple Summary or Abstract does not seem to be common. I recommend removing the statistical values in these sections. Additionally, the Simple Summary should clearly describe how the findings of this study are beneficial to society for a general audience. Please add this explanation.

Response 1: p-values have been removed from the simple summary, and a line has been added to describe how the results from this study can help society improve the care of cats.

Comment 2: Is it correct that there is a positive correlation between activity level and rainfall?

Response 2: This has been corrected

Comment 3: How do you expect seasonal and weather changes to affect cat behaviors? Do you predict that each have distinct impacts on cat behaviors? If there are any strengths in examining both seasonal and weather influences simultaneously, please make sure to highlight them.

Response 3: A line has been added to further clarify and support the choice to look at both the season and the weather (Lines 66 – 68).

Comment 4: Some of the figures contain the same images as earlier ones that were published, is a copyright okay?

Response 4: References to first publication have been added.

Comment 5: Could you provide more details about what each behavior specifically entails? For example, does "eating" also include drinking water? It would be helpful to include a clear definition of each behavior for better clarity.

Response 5: A table has been added with the definition of each behaviour (Table 1).

Comment 6: Please include the number of days per individual used in the final dataset, taking into account the excluded data.

Response 6: A table (Table 3) has been added showing the total number of days, hours, minutes and seconds included per season and in total in the final data analyses.

Comment 7: I recommend providing the results of the weight and temporal variations in weather as supplementary information, because they are not directly related to the main focus of the study.

Response 7: The results regarding the weather variables have been moved to supplementary information. Bodyweight, however, was kept in the results section, as it is an important part of the discussion.

Comment 8: The seasonal variation in behavior is significant. Although the weather in autumn and spring is similar, the seasonal differences are significant. However, the impact of varying rainfall between autumn and spring on cat behaviors is limited to a few behaviors. Could there be other factors influencing this, for example seasonal breeding patterns?

Response 8: In entire cats, breeding patterns can definitely play a role in behavioural changes, however, all cats in the current study were desexed, so this did not play a role. The effect of breeding, however, is now touched upon in a new paragraph added to the discussion (lines 500 – 517), along with some other examples.

Comment 9: As mentioned above, I recommend providing the data on weather variations as supplementary information, because it is not directly related to the main focus of the study.

Response 9: See comment and response 7.

Reviewer 2 Report

Comments and Suggestions for Authors

The article presents a longitudinal study investigating how seasonal and weather variations influence the behavior of domestic cats. The research was conducted over one year, using accelerometer data and machine learning models to track eight specific behaviors: active, eating, grooming, littering, lying, scratching, sitting, and standing. The research is interesting, but it isn't easy to extrapolate the results. Therefore, the number of animals is limited. I suggest that the authors take into account the following comments: 

1.- The study could benefit from references supporting the methodology used for classifying behaviors and describing clearly the criteria to consider each one. 

2.- Given that the article is mainly based on GLM, it could be acceptable to describe the model thoroughly and add which post hoc tests were used for pairwise comparisons to avoid confusion in statistical interpretation. Moreover, the statistical outcomes should be more explicitly discussed, explaining how data were analyzed and how the results were derived.

3.- It is essential that the discussion about seasonal changes affecting behavior be expanded to include more insights into the biological mechanisms behind these changes and how they might vary across individual cats.

4.- The horary in which the meteorological data were recorded was not indicated. Please add it.  

5.- Some data were eliminated to consider atypical variations. However, The fact that the number of animals evaluated was small could affect the decision and could be a bias in the interpretations.

Author Response

The article presents a longitudinal study investigating how seasonal and weather variations influence the behavior of domestic cats. The research was conducted over one year, using accelerometer data and machine learning models to track eight specific behaviors: active, eating, grooming, littering, lying, scratching, sitting, and standing. The research is interesting, but it isn't easy to extrapolate the results. Therefore, the number of animals is limited. I suggest that the authors take into account the following comments: 

Comment 1: The study could benefit from references supporting the methodology used for classifying behaviors and describing clearly the criteria to consider each one. 

Response 1: A new section has been added to the introduction to further support the use of accelerometers and machine learning for the classification of cat behaviour. As for the selection of behaviours included in the study: this has been outlined in more detail in the methods section.

Comment 2: Given that the article is mainly based on GLM, it could be acceptable to describe the model thoroughly and add which post hoc tests were used for pairwise comparisons to avoid confusion in statistical interpretation. Moreover, the statistical outcomes should be more explicitly discussed, explaining how data were analyzed and how the results were derived.

Response 2: The section on statistical analyses has been improved and now includes a mathematical form of the GLMM and the post hoc tests used has been added. The section has been slightly restructured, so it is now clearer what data and how the data were analysed.

Comment 3: It is essential that the discussion about seasonal changes affecting behavior be expanded to include more insights into the biological mechanisms behind these changes and how they might vary across individual cats.

Response 3: A new section on the impact of variables other than weather has been added to the discussion.

Comment 4: The horary in which the meteorological data were recorded was not indicated. Please add it.  

Response 4: Meteorological data were continuously collected on an hourly basis for the full duration of the study. This was stated in section 2.3, but has been further clarified, and dates for weather data collection have been added.

Comment 5: Some data were eliminated to consider atypical variations. However, the fact that the number of animals evaluated was small could affect the decision and could be a bias in the interpretations.

Response 5: Outliers were removed only when they resulted from disruptions unrelated to weather, as our primary interest was in the effect of weather on behaviour. Including these data points could have introduced confounding effects, making it more difficult to isolate weather-related behavioural changes. Given that individual variability was already accounted for in the GLMM, removing these specific outliers helped maintain the focus of the analysis without distorting natural behavioural variation. 

Reviewer 3 Report

Comments and Suggestions for Authors

This manuscript seeks to address if and how seasonal and weather variations influence domestic cat behaviours. I have no objections regarding how the research was conducted, as the methods are rigorous and the literature has been thoroughly explored, both in the introduction to the problem and in the discussion of the data. What leaves me perplexed, however, is the actual usefulness of the study. This is an investigation involving a very small number of animals (7 cats), all sterilized, with guaranteed food and shelter. It already seems almost miraculous that statistically significant results emerged under these conditions.

Furthermore, for the potential applicability of the findings in a different context, the authors have overlooked crucial aspects of domestic cats' lives—particularly the social context. No data were collected on social interactions of any kind, whether agonistic or affiliative. It is evident that the social environment influences an animal’s physical activity and sleep, just to mention aspects relevant to this study.

The only value I see in this work is methodological: it outlines a possible approach that could be applied in a more realistic setting. However, this point should be more explicitly stated in the paper.

Author Response

This manuscript seeks to address if and how seasonal and weather variations influence domestic cat behaviours. I have no objections regarding how the research was conducted, as the methods are rigorous and the literature has been thoroughly explored, both in the introduction to the problem and in the discussion of the data.

What leaves me perplexed, however, is the actual usefulness of the study. This is an investigation involving a very small number of animals (7 cats), all sterilized, with guaranteed food and shelter. It already seems almost miraculous that statistically significant results emerged under these conditions.

Response: Just because cats are housed in a similar environment, fed the same diets and have the same sexual status, we cannot assume their behaviour is the same. In previous studies, cat variability in activity was high. High cat variability has now been outlined in the introduction (Lines 102 – 103). The high resolution of the data in our study provides an opportunity to study cat behaviour in a degree of detail that was not previously available to us.

Furthermore, for the potential applicability of the findings in a different context, the authors have overlooked crucial aspects of domestic cats' lives—particularly the social context. No data were collected on social interactions of any kind, whether agonistic or affiliative. It is evident that the social environment influences an animal’s physical activity and sleep, just to mention aspects relevant to this study.

Response: When creating the machine learning models, we attempted to collect both agonistic and affiliative behaviour, however, neither of the two behaviours occurred frequently enough to provide a sample size big enough to be included in the model (see the validation paper: https://doi.org/10.3390/s23167165). We do agree that social interactions play an important role in behaviour, as we also found in a pet cat study (see https://doi.org/10.3390/s24082623). Here we found that cats in multi-cat households behave differently (less lying, more sitting, possible due to being more alert) than cats in single-cat households. We do want to iterate that the current study focused specifically about group-housed cats and we do not expect that the results could be extrapolated to cats living in a different situation. These points have been added into a new paragraph in the discussion (Lines 492 – 511).

The only value I see in this work is methodological: it outlines a possible approach that could be applied in a more realistic setting. However, this point should be more explicitly stated in the paper.

Response: It has been clarified that the results cannot be extrapolated to cats living under different conditions, and a sentence has been added to the discussion for the possibility of the method to be used in other settings.

Round 2

Reviewer 1 Report

Comments and Suggestions for Authors

This second version of the paper is a great improvement, the authors are to be commended.

Reviewer 2 Report

Comments and Suggestions for Authors

The authors improve the manuscript substantially